# State of Art and Perspectives in Catalytic Ozonation for Removal of Organic Pollutants in Water: Influence of Process and Operational Parameters

**Naghmeh Fallah** [1,*]**, Ermelinda Bloise** [1]**, Domenico Santoro** [2] **and Giuseppe Mele** [1]

[1] Department of Engineering for Innovation, University of Salento, Via Monteroni, 73100 Lecce, Italy
[2] Department of Chemical and Biochemical Engineering, University of Western Ontario, London, ON N6A 5B9, Canada
* Correspondence: naghmeh.fallah@unisalento.it

**Abstract:** The number of organic pollutants detected in water and wastewater is continuously increasing thus causing additional concerns about their impact on public and environmental health. Therefore, catalytic processes have gained interest as they can produce radicals able to degrade recalcitrant micropollutants. Specifically, catalytic ozonation has received considerable attention due to its ability to achieve advanced treatment performances at reduced ozone doses. This study surveys and summarizes the application of catalytic ozonation in water and wastewater treatment, paying attention to both homogeneous and heterogeneous catalysts. This review integrates bibliometric analysis using VOS viewer with systematic paper reviews, to obtain detailed summary tables where process and operational parameters relevant to catalytic ozonation are reported. New insights emerging from heterogeneous and homogenous catalytic ozonation applied to water and wastewater treatment for the removal of organic pollutants in water have emerged and are discussed in this paper. Finally, the activities of a variety of heterogeneous catalysts have been assessed using their chemical–physical parameters such as point of zero charge (PZC), pKa, and pH, which can determine the effect of the catalysts (positive or negative) on catalytic ozonation processes.

**Keywords:** catalytic ozonation; homogenous catalysts; heterogeneous catalysts; water treatment; VOSviewer; reaction mechanism

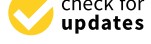

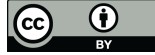

## 1. Introduction

Industrial wastewater has been under extensive research due to its hazardous effect on the aquatic, air, and soil environment, as well as human and animal health. Developing wastewater treatment technologies that are simple, safe, and efficient for the environment is becoming the twenty-first century's primary goal.

Advanced oxidation processes (AOPs) have shown great potential for the degradation and mineralization of recalcitrant and toxic organic pollutants compared to conventional treatment processes. This process is classified into two main general categories. The first category utilizes light energy such as ultraviolet (UV) light in conjunction with other chemical additives. There are processes under this category that associate other agents with UV, such as hydrogen peroxide (UV/$H_2O_2$), ozone (UV/$O_3$), titanium oxide (UV/$TiO_2$), and Fenton reagents (UV/Fenton). When no light source is used, the technology can be termed a dark oxidative process. Processes in this category include ozonation, Fenton's reagent, ultrasound, and microwaves. These processes are simultaneously based on the in situ generations of highly reactive transitory species ($H_2O_2$, $HO^{\bullet}$, $O^{2-}$, $O_3$) for the mineralization of refractory organic compounds and the inactivation of waterborne pathogens. Due to rapid oxidation reactions, AOPs are characterized by high reaction rates and short treatment times, which make them promising in wastewater treatment [1].

Ozone is a powerful oxidant with several advantages that make it an excellent material in the AOPs. There are many noticeable benefits of using $O_3$ in the wastewater treatment process. Benefits such as it rapidly reacting with bacteria, viruses, and protozoa, being efficient for organics degradation and inorganics removal, and removing color, taste, and odor. Although the ozonation process is a practical system, $O_3$ has low solubility and stability in water and a high production cost. To solve the mentioned disadvantages, some solutions have been explored, such as using fixed beds of porous glass or metals, solid catalysts, stirring, line mixers, contact towers, and an increase in retention time by large bubble columns or diffusers [1]. The combination of ozonation with other techniques is suggested as an intelligent solution. In this regard, various $O_3$-based AOPs, such as $OH^-/O_3$, $O_3/H_2O_2$, $O_3/UV$, $O_3/H_2O_2/UV$, $O_3/S_2O_8{}^{2-}$, $O_3$/biological treatment, and catalytic and photocatalytic ozonation, were introduced to the industry [2–7]. Each of these processes has its specific features and conditions.

In the $OH^-/O_3$ process, the pH value of the water matrices has a significant influence on both the direct ozonation efficiency and the generation of $HO^\bullet$ (indirect ozonation). At significantly high pH (pH > 8), the abundance of $OH^-$ can improve $HO^\bullet$ generation, which will enhance the ozonation of pollutants. However, high pH might cause the precipitation of calcium carbonate or other problems, which should be considered. In addition, the pH adjustment will increase the operational cost. In the so-called peroxone technique, $O_3$ and $H_2O_2$ would be combined [8]. The critical effect of combining $O_3$ and $H_2O_2$ is increasing oxidation efficiency. This occurs by converting $O_3$ to $HO^\bullet$ and improving $O_3$ transfer from the gas to the liquid phase [8]. The chemistry of the main reactions described above is shown in Equations (1)–(3).

$$H_2O_2 + O_3 \rightarrow HO^\bullet + HO_2^\bullet + O_2 \tag{1}$$

$$HO^\bullet + O_3 \rightarrow O_2 + HO_2^\bullet \tag{2}$$

$$O_3 + HO_2^\bullet \rightarrow 2O_2 + HO^\bullet \tag{3}$$

Another ozonation technique is the usage of ultraviolet light in combination with $O_3$ in an aqueous medium. This combination causes the increase in the $HO^\bullet$ formation and its concentration, consequently increasing the degradation efficiency. Equations (4) and (5) show that the formation of $H_2O_2$ as a by-product is possible, which will be degraded by the exact mechanism of $H_2O_2/UV$ [8], also increasing the treatment efficiency.

$$O_3 + H_2O + hv \rightarrow O_2 + 2HO^\bullet \tag{4}$$

$$2HO^\bullet \rightarrow H_2O_2 \tag{5}$$

The introduction of UV with $H_2O_2/O_3$ makes the previously mentioned techniques more efficient. This combination enhances $HO^\bullet$ generation and more efficiently allows the transformation of $H_2O_2$ to $HO^\bullet$ (Equation (6)) [9], consequently increasing the degradation rate.

$$O_3 + H_2O_2 + hv \rightarrow 2HO^\bullet + 3O_2 \tag{6}$$

Persulfate $S_2O_8{}^{2-}$ as a practicable material for water treatment would be combined with $O_3$. It is assumed that $O_3$ decomposes with the formation of $HO^\bullet$, which can then activate persulfate to generate $SO_4^{\bullet-}$ (Equation (7)) [10]. In turn, $SO_4^{\bullet-}$ can increase the formation of $HO^\bullet$, which leads to a multiradical system (Equations (8) and (9))

$$HO^\bullet + S_2O_8^{2-} \rightarrow SO_4^{\bullet-} + HSO_4^- + 1/2O_2 \tag{7}$$

$$SO_4^{\bullet-} + H_2O \rightarrow SO_4^{2-} + HO^\bullet + H^+ \tag{8}$$

$$SO_4^{\bullet-} + OH^- \rightarrow SO_4^{2-} + HO^\bullet \tag{9}$$

A popular ozonation process is an ozonation/biological treatment technology. This process can be divided into two types: (1) ozonation is used as pre-treatment, such as an $O_3$-biological activated carbon process, ozonation/batch aerobic biological system, and ozonation/aerated biological filter; (2) ozonation is used as post-treatment, such as membrane bioreactor/ozonation, activated-sludge biological treatment/ozonation, and sequencing batch biofilm reactor/ozonation. Since the intermediate products formed by ozonation and $O_3$-based processes are generally more biodegradable than their precursors, these intermediates can be much more easily removed by biological treatment processes. Therefore, if the water containing many inhibiting compounds is toxic to the biological cultures, in such cases, a biological treatment followed by pre-treatment ozonation is suitable for the application. On the other hand, if there are many biodegradable compounds, the pre-oxidation step obviously will only lead to the unnecessary consumption of chemicals. In this case, a biological pre-treatment followed by ozonation (removing non-biodegradable and toxic components with less oxidant consumption) may be more suitable [11].

In the ozonation process, the addition of some catalysts can promote the decomposition of the oxidant ($O_3$) to generate active free radicals, such as $HO^\bullet$. Compared with other $O_3$-based treatment methods, catalytic ozonation can reduce operational costs since it does not need additional energy costs such as for UV or for pH adjustment due to its effectiveness in a wide range of pH values. Moreover, the catalytic ozonation systems have shown exemplary performance in water treatment, with several advantages compared to ozonation alone. Several pieces of evidence based on published articles show that the catalytic ozonation process achieved higher mineralization of various organic compounds than the sole ozonation system [12]. All of these reasons make catalytic ozonation an interesting water treatment process and one of the main AOPs processes that received significant attention from scientists. Figure 1 shows the gradual increase in scientific publication since 2000 based on approximately 600 published articles found in the Web of Science collection database.

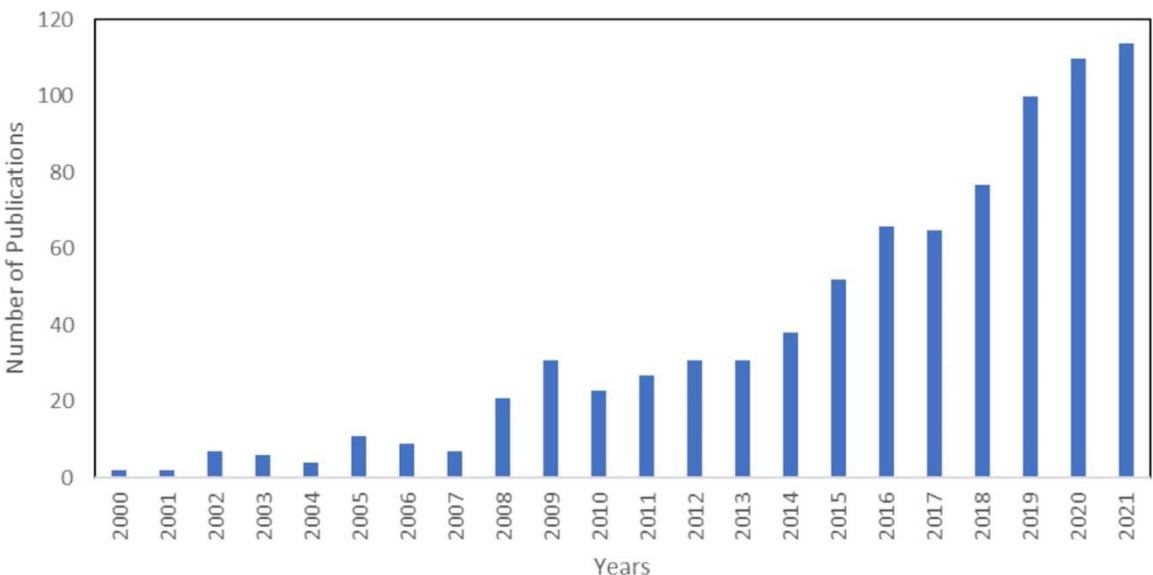

**Figure 1.** Published articles related to catalytic ozonation since 2000.

During these years, several catalysts have been proven to be effective in the enhancement of ozonation efficiency. Generally, the catalytic ozonation process can be divided into two types:

(1) Homogeneous catalytic ozonation, in which transition metal ions used as catalysts influence the rate of reaction, the selectivity of $O_3$ oxidation, and the efficiency of $O_3$ utilization. Two major mechanisms of homogeneous catalytic ozonation can be found: the $O_3$ decomposition by metal ions which generates free radicals; and the

complex formation between the metal ions and organic molecule following oxidation of the complex.

(2) Heterogeneous catalytic ozonation, which is based on the activation of $O_3$ to improve the ozonation of pollutants in the presence of a solid catalyst. Obviously, the key point in this process is to find the most appropriate catalyst, which is a solid material that in combination with $O_3$ shows a greater removal of a pollutant at a given pH value, compared to the separate processes of adsorption or ozonation alone. Among the most widely used catalysts in heterogeneous catalytic ozonation are metal/ bimetal/polymetal oxides, metal/metal oxides on supports, carbon-based materials, and the emerging category of multifunctional porous materials as metal–organic frameworks. The role of the catalyst in this process is to provide reaction sites for adsorption and catalysis. So, based on the interaction of catalysts with $O_3$ and micropollutants, three general major mechanisms can be found. (a) Adsorption of $O_3$ on the catalyst surface following $O_3$ decomposition to generate free radicals; (b) adsorption of micropollutants on the catalyst surface, then attacking by $O_3$ molecule; (c) adsorption of both $O_3$ and micropollutants on the catalyst and their reaction together.

Many factors may have an impact on the performance of the catalysts. The PZC value of the material, the acid/basic sites of the surface, the oxidative potential of the metals contained in the solid structure, the cation exchange capacity, the oxygen vacancies, etc., are examples of these factors [13–15]. An important research aspect is the specific role played by the point of zero charge (PZC) in the overall efficiency of the catalytic ozonation process. Several researchers measured the PZC value of their synthesized catalyst and explained their surface characteristics based on this factor [16–21]. On the other hand, there are studies that examined the role of oxygen vacancies in catalytic ozonation [14,22], and the effect of the lewis acid/basic sites of the surface [23–27]. Although there are some studies that prove that these two factors can also influence $O_3$ decomposition, there are a variety of papers in which authors do not report, nor discuss, the impact of these factors.

In the terms of process efficiency of catalytic ozonation, operational parameters and reaction mechanism in the process have a big impact. According to operational parameters, initial solution pH, $O_3$ dose, initial pollutant concentration, pressure, catalyst dosage, and temperature have major effects on efficiency. Our attention has been placed on the governing mechanisms of the process based on the chemical properties of catalysts, physical properties of catalysts, natural properties of target pollutants (pKa), and pH of the solution as described in this article.

## 2. A Bibliometric Analysis Using VOSviewer

The scientific articles about catalytic ozonation published between 2000 and 2021 were scanned in the Web of Science (WOS) collection database. The words "Catalytic Ozonation" were used as the keywords to achieve the relevant publications. VOS viewer was applied to perform the bibliometric analysis of these articles. In this respect, 600 publications on the topic of catalytic ozonation were identified in the WOS core database.

Bibliometric analysis of the keywords in publications was studied. In this respect, all provided keywords in the articles related to catalytic ozonation that occurred more than 20 times in the WOS core database were enrolled in the final analysis. Based on the catalytic ozonation articles in the English language, of the 1441 keywords accrued, 26 of these keywords had appeared 20 times more frequently than the others. The keyword "catalytic ozonation" was the most frequently occurring one, with an occurrence of 197 and total link strength of 177. Following the previously mentioned keyword, "ozone", "degradation", and "oxidation" occurred 169, 154, and 129 times, respectively. Figure 2 illustrated the bibliometric analysis and VOSviewer visualization of the keywords in articles related to catalytic ozonation.

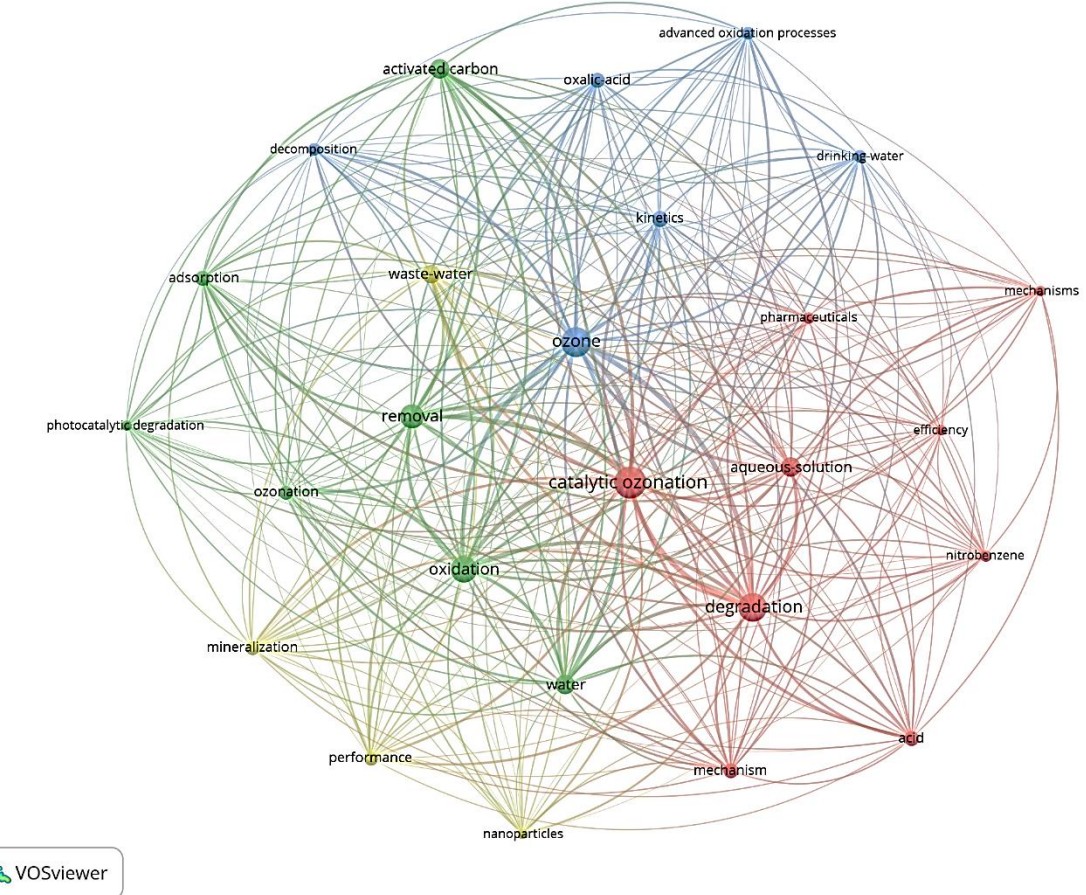

**Figure 2.** Co-occurrence of the keywords in articles related to catalytic ozonation visualized by VOSviewer software.

Based on the WOS collection database results in the case of 'catalytic ozonation', a bibliometric analysis of the co-authorship between countries was studied. All countries that published more than five articles in the WOS core database related to catalytic ozonation were enrolled in the final analysis.

In this respect, the top ten most active countries in the field of catalytic ozonation based on the number of citations, publications, and total link strength, are listed in Table 1. Of the 47 countries that worked on catalytic ozonation topics, 14 countries had more than 5 published articles. Based on the information gathered, the country most active in the field of catalytic ozonation is "China", with 185 publications, 422 citations, and a total link strength of 20. Following that, Iran and Canada take second and third places, respectively.

All these bibliometric analyses show the importance of catalytic ozonation among scientists all around the world. In the absence of a comprehensive and precise survey about the developments in catalytic ozonation processes, it would not be easy to propose novel investigations to optimize water treatment performance in terms of mineralization, industrial application, and economic capability. In light of these considerations, this study aims to recognize the popular and leading articles in catalytic ozonation, summarizing new visions on the evaluation of both heterogeneous and homogenous processes for the degradation and mineralization of various toxic organic pollutants in water.

**Table 1.** The top 10 active countries in the field of catalytic ozonation.

| Rank | Country | Number of Citations | Number of Documents | Total Link Strength |
|---|---|---|---|---|
| 1 | China | 422 | 185 | 20 |
| 2 | Iran | 82 | 37 | 4 |
| 3 | Canada | 38 | 14 | 5 |
| 4 | USA | 41 | 13 | 8 |
| 5 | Pakistan | 28 | 12 | 8 |
| 6 | Brazil | 22 | 8 | 1 |
| 7 | Turkey | 25 | 8 | 1 |
| 8 | France | 23 | 8 | 0 |
| 9 | Australia | 10 | 7 | 5 |
| 10 | Greece | 14 | 7 | 1 |

## 3. Homogeneous Catalytic Ozonation

Transition metals such as Fe(II), Fe(III), Mn(II), Ni(II), Co(II), Pb(II), and Zn(II) used as homogenous catalysts in the aqueous solution are commonly involved in two mechanistic steps of the ozonation processes: initiation of the $O_3$ decomposition reaction followed by the generation of the hydroxyl radicals, and oxidation reaction between catalyst and organic pollutants. Some homogeneous catalytic systems selected among the most influencing articles reported in the current literature are listed in Table 2. Details regarding the kind of used catalysts, target pollutants, operating conditions, and removal results are included in this table.

**Table 2.** Literature reports on different homogenous catalysts in the ozonation process.

| Target Pollutants | Catalysts | Operating Conditions | Removal Results | Ref. |
|---|---|---|---|---|
| Aniline | Fe(II) | $O_3$ dose: 0.5 g/h; Catalyst dose: 1 mmol/dm$^3$; pH: 3.3; T: 25 °C; t: 15 min | 132 TOC removal | [28] |
| 4-chlorophenol | Fe(II) | $O_3$ dose: 0.5 g/h; Catalyst dose: 1 mmol/dm$^3$; pH: 3.3; T: 25 °C; t: 15 min | 144 TOC removal | [28] |
| Oxalic acid (OA) | Fe(III) | $O_3$ dose: 8.2 mg/L; Catalyst dose: 1 mg/L; pH: 2; T: 20 °C; t: 3 h | 7% OA removal | [29] |
| 1,3,6-naphthalenetrisulfonic acid (NTS) | Fe(II) | $O_3$ dose: $1.04 \times 10^{-4}$ mol/dm$^{-3}$; Catalyst dose: $1.25 \times 10^{-4}$ mol/dm$^{-3}$; pH: 2; T: 25 °C; t: 30 min | 79% NTS degradation | [30] |
| Lipid | Fe(II) | $O_3$ dose: 0.6 g/L; Catalyst dose: 7 mg/L; pH: 6.75; T: 25 °C; t: 60 min | 96.7% lipid degradation | [31] |
| Chlorobenzenes | Fe(II) | $O_3$ dose: 1.5 g $O_3$/TOC; Catalyst dose: $6 \times 10^{-5}$ mol/L; pH: 7; t: 20 min | 55% COD removal | [32] |
| Chlorobenzenes | Fe(III) | $O_3$ dose: 1.5 g $O_3$/TOC; Catalyst dose: $6 \times 10^{-5}$ mol/L; pH: 7; t: 20 min | 12% COD removal | [32] |
| Aniline aerofloat (AAF) | Fe(II) | $O_3$ dose: 2.08 mg/min. L; Catalyst dose: 10 mg/L; pH: 8; T: 25 °C; t: 180 min | 80% COD removal | [33] |
| AAF | Fe(III) | $O_3$ dose: 2.08 mg/min. L; Catalyst dose: 10 mg/L; pH: 8; T: 25 °C; t: 180 min | 76% COD removal | [33] |
| C.I. Reactive Red 2 (RR2) | Fe(III) | $O_3$ dose: 200 mL/min; Catalyst dose: 0.6 mM; pH: 2; T:NR; t: 6 min | 1.278 of decolorization rates (1/min) | [34] |
| RR2 | Fe(II) | $O_3$ dose: 200 mL/min; Catalyst dose: 0.6 mM; pH: 2; T:NR; t: 6 min | 1.299 of decolorization rates (1/min) | [34] |
| p-Chlorobenzoic acid (p-CBA) | Fe(II) | $O_3$ dose: 2 mg/L; Catalyst dose: 1 mg/L; pH: 7; T: 23 °C; t: 15 min | 92.5% p-CBA degradation | [35] |
| RR2 | Zn(II) | $O_3$ dose: 200 mL/min; Catalyst dose: 0.6 mM; pH: 2; T:NR; t: 6 min | 1.015 of decolorization rates (1/min) | [34] |

**Table 2.** *Cont.*

| Target Pollutants | Catalysts | Operating Conditions | Removal Results | Ref. |
|---|---|---|---|---|
| p-CBA | Co(II) | $O_3$ dose: 2 mg/L; Catalyst dose: 1 mg/L; pH: 7; T: 23 °C; t: 15 min | 95.5% p-CBA degradation | [35] |
| RR2 | Co(II) | $O_3$ dose: 200 mL/min; Catalyst dose: 0.6 mM; pH: 2; T:NR; t: 6 min | 0.843 of decolorization rates (1/min) | [34] |
| OA | Co(II) | $O_3$ dose: 30 mg/L; Catalyst dose: 0.8 mg/L; pH: 2.5; T:NR; t: 90 min | 70% OA removal | [36] |
| OA | Co(II) | $O_3$ dose: $5 \times 10^{-3}$ mol/L; Catalyst dose: 4 mg/L; pH: 2.5; T: 25 °C; t: 30 min | 99.3% TOC removal | [37] |
| Formic acid | Co(II) | $O_3$ dose: $5 \times 10^{-3}$ mol/L; Catalyst dose: 4 mg/L; pH: 2.5; T: 25 °C; t: 35 min | 60.2% TOC removal | [37] |
| Carboxylic acids | Cu(II) | $O_3$ dose: 71 mg/L; Catalyst dose: 20 µg/L; pH: natural; T:NR; t:NR | 75% TOC reduction | [38] |
| AAF | Cu(II) | $O_3$ dose: 2.08 mg/min. L; Catalyst dose: 10 mg/L; pH: 8; T: 25 °C; t: 180 min | 75% COD removal | [33] |
| N-dimethylpropyl-2-pyrrolidone (NDPP) | Pd(II) | $O_3$ dose: 250 mL/min. L; Catalyst dose: $9.4 \times 10^{-5}$ M; pH: 2; T: 25 °C; t: 30 min | 73% NDPP removal | [39] |
| RR2 | Ni(II) | $O_3$ dose: 200 mL/min; Catalyst dose: 0.6 mM; pH: 2; T:NR; t: 6 min | 0.822 of decolorization rates (1/min) | [34] |
| RR2 | Mn(II) | $O_3$ dose: 200 mL/min; Catalyst dose: 0.6 mM; pH: 2; T:NR; t: 6 min | 3.295 of decolorization rates (1/min) | [34] |
| Pyruvic acid | Mn(IV) | $O_3$ dose: 0.5 g/h; Catalyst dose: 200 mg; pH: 3; T: 25 °C; t: 60 min | | [40] |
| Atrazine (ATZ) | Mn(II) | $O_3$ dose: 2.54 mg/L; Catalyst dose: 0.3 mg/L; pH: 7; T: 21 °C; t: 4 min | 96% ATZ removal | [41] |
| ATZ | Mn(II) | $O_3$ dose: 2.28 mg/L; Catalyst dose: 1 mg/L; pH: 7; T: 23 °C; t: 5 min | 70% ATZ removal | [42] |
| ATZ | Mn(IV) | $O_3$ dose: 2.28 mg/L; Catalyst dose: 1 mg/L; pH: 7; T: 23 °C; t: 5 min | 37% ATZ removal | [42] |
| 2,4-dinitrotoluene (DNT) | Mn(II) | $O_3$ dose: 5.6 mg/L; Catalyst dose: 0.2 mg/L $Mn^{2+}$ and 4 mg/L OA; pH: 5.5; T: 25 °C; t: 15 min | 65% DNT removal | [43] |
| 2,4-dichlorophenol | Mn(II) | $O_3$ dose: 8.4 mg/L; Catalyst dose: 0.5 mg/L; pH: 5.5; T: 25 °C; t: 30 min | 80% TOC removal | [44] |
| NTS | Mn(II) | $O_3$ dose: $1.04 \times 10^{-4}$ mol/dm$^{-3}$; Catalyst dose: $1.25 \times 10^{-4}$ mol/dm$^{-3}$; pH: 2; T: 25 °C; t: 30 min | 72% NTS degradation rate | [30] |
| OA | Mn(II) | $O_3$ dose: $5 \times 10^{-3}$ mol/L; Catalyst dose: 4 mg/L; pH: 2.5; T: 25 °C; t: 30 min | 82% TOC removal | [37] |
| Formic acid | Mn(II) | $O_3$ dose: $5 \times 10^{-3}$ mol/L; Catalyst dose: 4 mg/L; pH: 2.5; T: 25 °C; t: 35 min | 61% TOC removal | [37] |
| Chlorobenzenes | Mn(II) | $O_3$ dose: 1.5 g $O_3$/TOC; Catalyst dose: $6 \times 10^{-5}$ mol/L; pH: 7; T:NR; t: 20 min | 66% COD removal | [32] |
| Lipid | Mn(II) | $O_3$ dose: 0.6 g/L; Catalyst dose: 3 mg/L; pH: 6.75; T: 25 °C; t: 60 min | 93% lipid degradation | [31] |
| Simazine | Mn(II) | $O_3$ dose: 9.5 mg/L; Catalyst dose: 0.2 mg/L; pH: 7; T: 25 °C; t: 30 min | 90% Simazine conversion | [45] |

NR-value not reported, TOC total organic carbon, COD chemical oxygen demand.

Different kinds of transitional metals are used as homogeneous catalysts however, Fe(II) and Mn(II) have been reported to be the most efficient catalysts for water purification purposes. This trend in the research has underlying scientific logic. By focusing on the general features of these transition metals' chemistry, their wide range of oxidation states, and complex ion formation, this popularity for using them as a catalyst makes sense. Mn with an atomic number of 25 has the highest number of unpaired electrons in the d-subshell, and it shows variable oxidation states in its compounds such as (II) in $Mn^{2+}$, (III) in $Mn_2O_3$, (IV) in $MnO_2$, (VI) in $MnO_4^{2-}$, and (VII) in $MnO^{4-}$. Fe has two standard oxidation states, $Fe^{2+}$ and $Fe^{3+}$, and a less common (VI) oxidation state in $FeO_4^{2-}$. Existing unpaired electrons and vacant orbitals in these transition metal structures enable them to accept electrons from other ions of molecules to form complex compounds. So this ability causes the adsorption of other substances onto their surface and activates them in the pro-

cess, which is the objective function of a catalyst. The application of these two important transitional metals is explained in the following publications as examples.

Ramos et al. [31] evaluated the efficiency of the Fe(II) catalyst in the ozonation process for lipid degradation. Milk was chosen as the lipid source. It is observed that under neutral conditions, low catalyst dosages are enough to cause the almost complete degradation of lipids (96.7%). Fu et al. [33] investigated the homogeneous catalytic ozonation of AAF collector by coexisting transition metallic ions (Fe(II), Fe(III), Cu(II), Pb(II), and Zn(II)) in flotation wastewaters. Based on this research, the following order of the degradation rate was achieved: $O_3$/Fe(II) > $O_3$/Fe(III) > $O_3$/Cu(II) > $O_3$/Pb(II) > $O_3$/Zn(II) ≈ $O_3$-alone. The best catalytic activity gained by Fe(II) had a 31.15% growth of degradation rate and achieved an increase of 42.26% for the AAF mineralization compared to $O_3$-alone. Xiao et al. [44] studied the mineralization of DCP in the ozonation process with Mn(II) as a catalyst. This study suggested that in the optimal condition of 0.5 (mg/L) catalyst dose and pH: 5.5, Mn(II) catalytic ozonation had a strong ability to degrade DCP and had 80% TOC removal in water solution.

In terms of the popularity of using this process, it is worth noting that most of the studies related to homogenous catalytic ozonation belong to the first decade of the 20th century. As can be observed in the table, the most significant disadvantage is that this catalytic process is mainly carried out in acidic pH values and not near the natural pH. At the same time, micropollutants, mostly emergent organic pollutants, usually exist in wastewater at the pH range of 6–8. Noting that although homogeneous catalytic ozonation processes can effectively improve the removal of organic contaminants in water in some cases, the addition of metal ions might result in secondary pollution, which causes limiting of their application.

However, in previous years, some scientists showed interest in using transitional metals for the catalytic ozonation process, working at the natural pH and real wastewater. Furthermore, some scientists solved the drawback of introducing these harmful metal ions in the aqueous environment by presenting the idea that some of these transition metallic ions usually coexisted in real wastewater. Several studies confirmed that $Fe^{2+}$, $Cu^{2+}$, $Zn^{2+}$, $Pb^{2+}$, and $Co^{2+}$, usually coexist with flotation reagents in the flotation pulp because of the dissolution of minerals or in the bastnaesite flotation pulp, some transition metal ions such as $Fe^{2+}$, $Cu^{2+}$, $Zn^{2+}$, and $Pb^{2+}$ were determined [46,47]. Finally, Fu and colleagues [33] showed that these coexisting transition metallic ions can be used as in situ catalysts. So, it can be deduced that this process can have pleasing prospects by considering some improvements in the future.

## 4. Heterogeneous Catalytic Ozonation

Catalysts in solid form with high stability and efficiency were widely studied in catalytic ozonation systems. In this section, the activity and efficiency of heterogeneous catalysts in the ozonation process have been evaluated by focusing our attention on essential parameters such as:

- Chemical properties of catalysts: crystallographic and morphological, chemical stability.
- Physical properties of catalysts: point of zero charge (PZC), mechanical strength, surface area, pore volume, and porosity.
- Natural properties of target pollutants (pKa) and pH of the solution.

In each category of catalysts, these parameters are different to achieve the optimum efficiency in the water treatment process. Previous review articles [1,15,48] mentioned that the mechanism of catalytic ozonation is too complicated due to the contradictory catalytic mechanisms proposed by different research groups. In this article, by scrutinizing the governed mechanism of the process based on their conditions, this lack of understanding would have new interpretations.

Generally, the interaction between catalyst, pollutant, and $O_3$ determines the governing mechanism of this process. Moreover, each of these active components' behaviors depends on other factors and some of these have more influence than others. In the follow-

ing, possible conditions based on the most influencing factors for each active component are expressed, and the governing mechanism in each specific condition is discussed.

Considering that the PZC is generally described as the pH value at which the net charge of the catalyst's surface is equal to zero.

The positively charged surface catalyst could be found under three different conditions which are proposed in Figure 3.

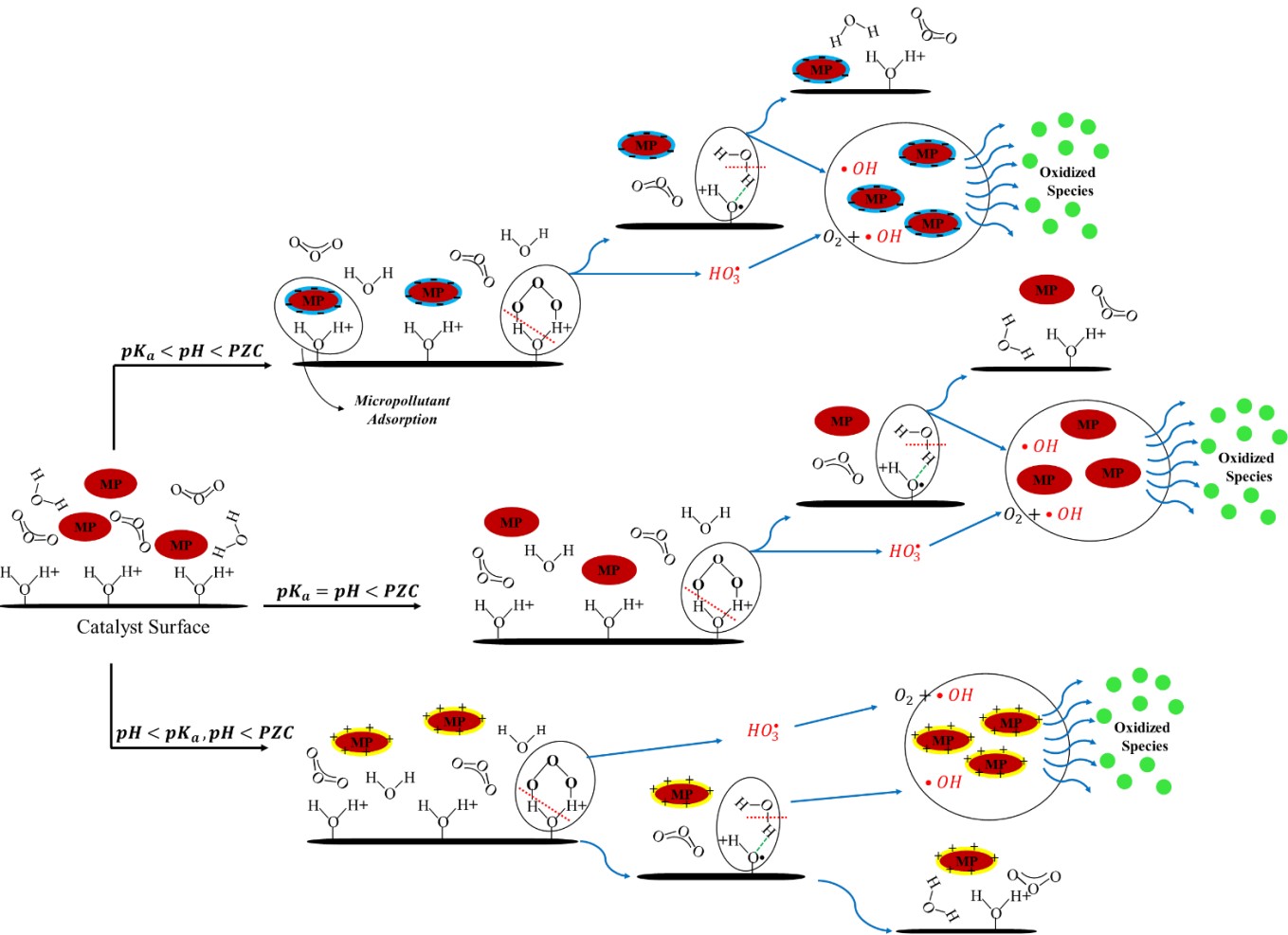

**Figure 3.** Proposed three influencing mechanisms using positively charged catalysts.

$pK_a < pH < PZC$ : Catalyst is positively charged, pollutant is negatively charged.

In this condition, the negatively charged micropollutants can be adsorbed on the positively charged catalyst surface. Thus, the contaminants would be close to the area where the generation of HO$^\bullet$ radicals happens, which means HO$^\bullet$ can quickly oxidize them in the environment. On the other hand, by adsorbing the micropollutants on the catalyst surface, the active area for adsorption of $O_3$ will be limited, which has a negative effect on the HO$^\bullet$ generation in the environment.

$pH < PZC$ , $pH = pK_a$ : Catalyst is positively charged, pollutant is uncharged.

There is no effective interaction between the catalyst and micropollutant in this condition, so the generation of HO$^\bullet$ by $O_3$ decomposition is the only effective parameter here.

$pH < PZC$ , $pH < pK_a$ : Catalyst and pollutant are positively charged.

In these conditions, the HO$^\bullet$ radicals, which are highly useful in the oxidation of micropollutants, would be generated in a short time with the adsorption of $O_3$ on the catalyst surface. Adsorption of $O_3$ molecules by hydroxyl radical on the surface causes a

generation of intermediate species ($OH_3^\bullet$) and another radical on the catalyst surface. The produced intermediate ($OH_3^\bullet$) turns into reactive $HO^\bullet$ radicals and $O_2$ in an in situ reaction. In parallel, the radical species on the catalyst surface would adsorb water molecules and produce reactive $HO^\bullet$ radicals. Due to the non-selectively behavior of $HO^\bullet$, it can oxidize almost all organic contaminants, which causes the high removal efficiency of micropollutants. Furthermore, based on the charge of micropollutants ($pK_a$), pulling of micropollutants or expelling of micropollutants on the catalyst surface might happen, and each of these conditions can affect removal efficiency. In this condition, the desorption of micropollutants from the catalyst would happen due to the repulsive electrostatic forces, which means the contaminants do not occupy the active surface sites, and pore blocking and associated fouling on the surface would be limited.

There are conditions that the catalyst is uncharged. Uncharged catalysts usually have hydroxyl radicals on the surface. Although these radicals are uncharged, they can be the starter part for $O_3$ decomposition. In the $O_3$ decomposition reaction chain, after the adsorption of $O_3$ on the surface, chemical bond stretching and breaking can happen in several ways. In one case, after bond breaking, $HO^\bullet$ and $O_2$ would be generated directly. In another case, some intermediate species such as $OH_3^\bullet$ and $O_3^-$ would be produced. The produced intermediate ($OH_3^\bullet$) turns into reactive $HO^\bullet$ radicals and $O_2$ in an in situ reaction. In parallel, $O_3^-$ causes other chain reactions, which finally produce $HO^\bullet$. Figure 4 proposed influencing mechanisms when the catalyst is uncharged.

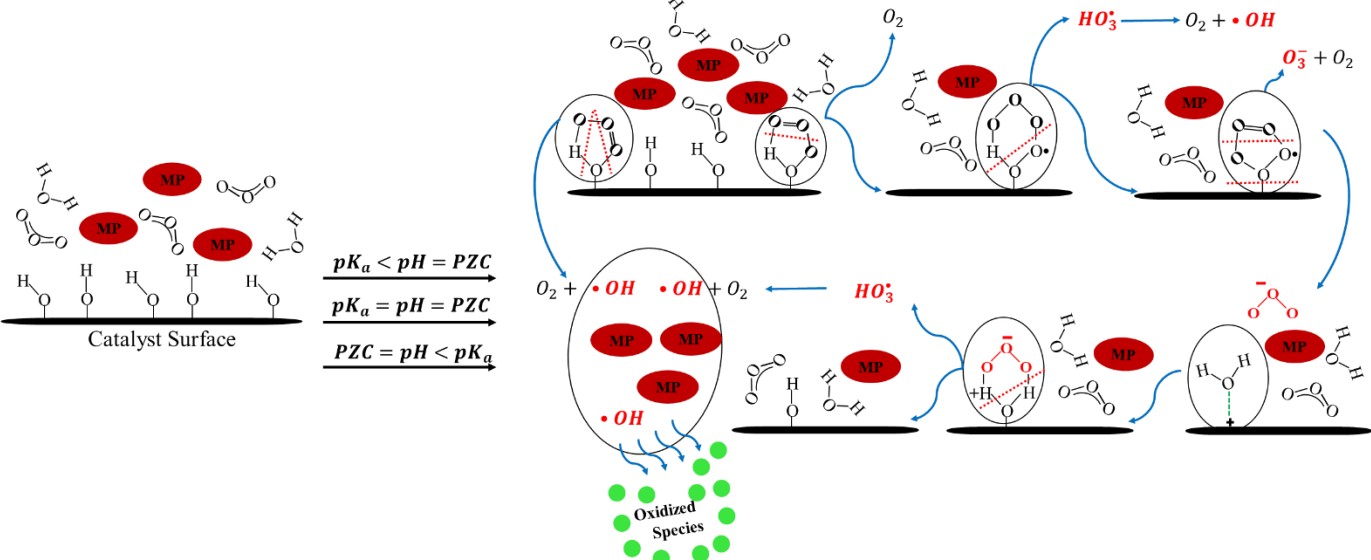

**Figure 4.** Proposed three influencing mechanisms using uncharged catalysts.

$pK_a < pH$, $PZC = pH$ : Catalyst is uncharged, Pollutant is negatively charged.
$pH < pK_a$ , $PZC = pH$ : Catalyst is uncharged, Pollutant is positively charged.
$PZC = pH = pK_a$ : Catalyst is uncharged, Pollutant is uncharged.

Due to the uncharged catalyst surface, there is no considerable difference between the three conditions in this subcategory.

Figure 5 summarizes three conditions related to the negatively charged surface of the catalyst. This situation seems to be the most favored for the adsorption and subsequent decomposition of $O_3$.

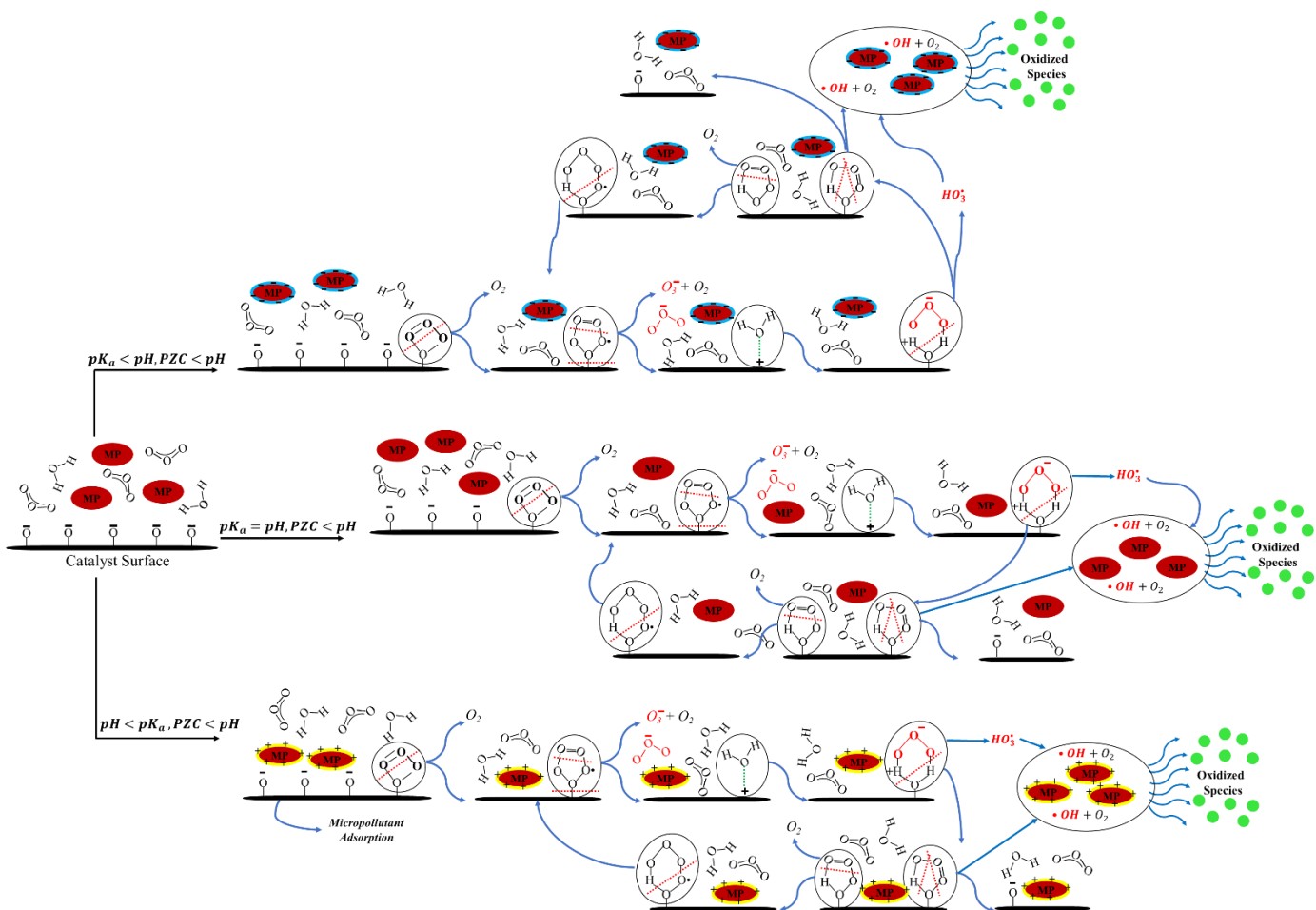

**Figure 5.** Proposed three influencing mechanisms using negatively charged catalysts.

In fact, the adsorption of $O_3$ on a surface would generate intermediate species such as $O_3^-$ and $OH_3^\bullet$, considered as precursors of the high oxidant species $HO^\bullet$.

$PZC < pH < pK_a$ : Catalyst is negatively charged, pollutant is positively charged.

Although in this condition, adsorption of $O_3$ is favored, adsorption of the pollutant on the catalyst surface can occur too. Thus, the pore blocking and associated fouling on the surface and reducing the active surface sites might have a negative effect on the $HO^\bullet$ generation in the environment. On the other side, closing to the area of $HO^\bullet$ radicals generation might cause quickly oxidize of micropollutants.

$PZC < pH$, $pH = pK_a$ : Catalyst is negatively charged, pollutant is uncharged.

There is no effective interaction between the catalyst and micropollutants in this condition, so the generation of $HO^\bullet$ by $O_3$ decomposition is the only effective parameter here.

$PZC < pH$ , $pK_a < pH$ : Catalyst and pollutant negatively charged.

In this condition, the negatively charged micropollutants would be repulsed from the negatively charged catalyst surface due to the electrostatic forces. Thus, the micropollutants do not occupy the active surface sites.

The values of pKa for different pollutants reported in Table 3 have been examined in order to compare and better evaluate the fundamental mechanisms proposed in this section.

**Table 3.** Values of pKa for different pollutants.

| Target Pollutants | pKa | Ref. | Target Pollutants | pKa | Ref. |
|---|---|---|---|---|---|
| Isoniazid | 1.82 | [49] | Ciprofloxacin (CPF) | 6.38 | [50] |
| Oxalic acid (OA) | pKa1 = 1.14; pKa2 = 3.64 | [51] | 4-nitrophenol | 7.15 | [52] |
| Phenacetin | 2.2 | [53] | Fluoxetine | 8.7 | [50] |
| Amoxicillin | 2.4 | [50] | Atenolol | 9.16 | [50] |
| Humic acids | 2.5 | [54] | Acetaminophen | 9.38 | [55] |
| Sulfamethazine | 2.65 | [50] | 4-Chloro phenol | 9.41 | [52] |
| Salicylic acid | 3.5 | [50] | Paracetamol (PCT) | 9.39–9.5 | [50,56] |
| Methylene blue | 3.8 | [57] | 4-Chloro-2-methyl | 9.71 | [52] |
| Reactive black-5 | 3.8 | [58] | Phenol (PH) | 9.98 | [59] |
| Furosemide | 3.9 | [50] | m-cresol | 10.1 | [60] |
| Diclofenac | 4.15 | [50] | Bisphenol-A (BPA) | 10.29 | [61] |
| Naproxen | 4.2 | [50] | 2,4-dimethylphenol | 10.4 | [52] |
| Ibuprofen | 4.51 | [50] | Orange (II) | 11.4 | [62] |
| Acetic acid | 4.76 | [63] | RR189 | 11.7 | [64] |
| (SMX) | 5.6–5.8 | [65,66] | Carbamazepine | 13.9 | [50] |
| Naphthenic acid | 5–6 | [67] | | | |

The PZC values for selected catalysts referring to the reviewed articles are shown in Table 4. Based on the type of catalysts, the pathway of the catalytic ozonation can also be varied. Although some catalysts exhibit PZC values in narrow ranges due to various impurity contents, synthesis routes, or thermal histories, this difference appears to be of little relevance to their catalytic properties.

**Table 4.** Range of PZC for selected catalysts.

| Catalysts | PZC | Ref. | Catalysts | PZC | Ref. |
|---|---|---|---|---|---|
| $SiO_2$ | 2.6 | [68,69] | $CoFe_2O_4$ | 7.31 | [70] |
| $MnO_2$ | 3–5 | [1,71] | $Fe_3O_4$ nanoparticles | 7.4 | [72] |
| MWCNTs | 4.2 | [68] | $CeO_2$ | 8.1 | [73] |
| $\alpha\text{-}Al_2O_3$ | 4.2 | [74] | $\gamma\text{-}Al_2O_3$ | 8.3–8.9 | [75,76] |
| AC | 4.9 | [68] | NiO | 8.45 | [77] |
| $NiCo_2O_4$ | 5 | [78] | ZnO | 9 | [1] |
| FeOOH | 5.9 | [79] | $\alpha\text{-}Al_2O_3$ | 9.4 | [75] |
| $Ce_3O_4$ | 5–8 | [69,80] | $MgFe_2O_4$ | 9.8 | [81] |
| $CuFe_2O_4$ | 6–7 | [53] | CuO | 10 | [82] |
| $TiO_2$ | 6.2–6.6 | [1] | MgO | 12–13 | [83] |
| $Fe_2O_3$ | 6–9 | [84] | | | |

The main catalyst types applied in heterogeneous catalytic ozonation are metal/bimetal/polymetal oxides, metals or metal oxides on supports, and carbon-based materials. In each of these categories, the most popular catalysts and published works in recent years are summarized in this review.

### 4.1. Metal/Bimetal/Polymetal Oxides

Several metal oxides have been introduced to promote the heterogeneous catalytic ozonation processes. Some of these metal oxides are more popular than others in the catalytic ozonation process, such as $Al_2O_3$, MgO, $CeO_2$, $MnO_2$, NiO, ZnO, etc. Furthermore, some bimetal/polymetal oxides were widely applied due to their high stability as well as high catalytic activity, such as $CuFe_2O_4$, Mn-Ce-O, $ZnAl_2O_4$, etc. Table 5 compiles the literature results employing metal/bimetal/polymetal oxides for the degradation of pollutants in the catalytic ozonation process. As can be seen, operating conditions (pH, $O_3$ dosage, catalyst dose, etc.), the kind of used catalysts, target pollutants, and removal results are reported. The articles for summarizing in this part were chosen according to their high citations, and newness.



**Table 5.** Literature reports on different metal/bimetal/polymetal oxides as catalysts in the ozonation process (see Figures 3–5, respectively).

| Catalysts | Target Pollutants | Operating Conditions | | Removal Results | Year | Ref. |
|---|---|---|---|---|---|---|
| | | pH < PZC | pH > pK$_a$, pH ≈ pK$_a$, pH < pK$_a$ | | | |
| γ-Al$_2$O$_3$ PZC = 8.9 + | Ibuprofen pK$_a$ = 4.51 − | O$_3$ dose: 0.5 mg/min; Catalyst dose: 5 g; pH: 7.2; T: 20 °C; t: 30 min | | 83% ibuprofen removal | 2015 | [76] |
| CoFe$_2$O$_4$ PZC = 7.31 + | OA pKa1 = 1.14; pKa2 = 3.64 | O$_3$ dose: 14 ± 1 mg/L; Catalyst dose: 1 g/L; pH: 2.3; T: NR; t: 120 min | | 68.3% TOC removal | 2017 | [70] |
| γ-Al$_2$O$_3$ PZC = 8.9 + | Cumene | O$_3$ dose: 0.5 mg/min; Catalyst dose: 5 g; pH: 7.2; T: 20 °C; t: 30 min | | 58% cumene removal | 2015 | [76] |
| MgO PZC = 12–13 + | Methylene blue pK$_a$ = 3.8 − | O$_3$ dose 5 mg/L; Catalyst dose: NR; pH: 9; T: NR; t: 60 min | | 50% COD removal | 2016 | [85] |
| γ-Al$_2$O$_3$ PZC = 8.9 + | 1,2-dichlorobenzene | O$_3$ dose: 0.5 mg/min; Catalyst dose: 5 g; pH: 7.2; T: 20 °C; t: 30 min | | 45% 1,2 dichlorobenzene removal | 2015 | [76] |
| γ-Al$_2$O$_3$ PZC = 8.9 + | Acetic acid pK$_a$ = 4.76 − | O$_3$ dose: 0.5 mg/min; Catalyst dose: 5 g; pH: 7.2; T: 20 °C; t: 30 min | | 19% acetic acid removal | 2015 | [76] |
| Ce-O PZC = 8.5 + | CI Reactive Blue 5 (RB5) | O$_3$ dose: 50 g/Nm$^3$; Catalyst dose: 350 mg; pH: 5.6; T: 25 °C; t: 3 h | | 85% TOC removal | 2009 | [86] |
| γ-Al$_2$O$_3$ PZC = 8.3 + | Petroleum refinery wastewater | O$_3$ dose: 5 mg/min; Catalyst dose: 0.5 g; pH: 8.15; T: 30 °C; t: 40 min | | 45.9% COD removal | 2017 | [87] |
| MgO PZC = 12–13 + | Acetaminophen pK$_a$ = 9.38 + | O$_3$ dose: 50 mg/L; Catalyst dose: 2 g/L; pH: 5.4; T: NR; t: 10 min | | 100% ACT degradation | 2017 | [16] |
| MgO PZC = 12–13 + | RR198 pK$_a$ = 11.7 + | O$_3$ dose: 0.2 g/h; Catalyst dose: 5 g/L; pH: 8; T: 23 °C; t: 9 min | | 100% RR198 removal | 2009 | [88] |
| MgO PZC = 12–13 + | 4-Chloro phenol pK$_a$ = 9.41 + | O$_3$ dose: 2.5 mg/min; Catalyst dose: 1.0 g/L; pH: 6.2; T: NR; t: NR | | 99.5% removal efficiency | 2015 | [17] |
| β-FeOOH PZC = 5.9 + | 4- Chloro phenol pK$_a$ = 9.41 + | O$_3$ dose: 28.24 mg/L; Catalyst dose: 1 g/L; pH: 3.5; T: NR; t: 40 min | | 99% removal efficiency | 2015 | [89] |
| γ-Al$_2$O$_3$ PZC = 8.3–8.9 + | PCT pK$_a$ = 9.39–9.5 + | O$_3$ dose: 3 mg/min; Catalyst dose: 5 mg/L; pH: 7; T:NR; t: 9 min | | 98% PCT removal | 2018 | [90] |
| MgO PZC = 12–13 + | PH pK$_a$ = 9.98 + | O$_3$ dose: 0.25 g/h; Catalyst dose: 4 g/L; pH: 7; T: 25 °C; t: 80 min | | 96% PH removal, 70% COD removal | 2010 | [91] |
| γ-Al$_2$O$_3$ PZC = 8.3–8.9 + | Fluoxetine pK$_a$ = 8.7 + | O$_3$ dose: 30 mg/L; Catalyst dose: 1 g/L; pH: 7; T: 25 °C; t: 17 min | | 86% Fluoxtenine removal | 2019 | [92] |
| MgFe$_2$O$_4$ PZC = 9.8 + | Acid Orange II pK$_a$ = 11.4 + | O$_3$ dose: 5 mg/L; Catalyst dose: 0.5 mmol/L; pH: 4.6–9.6; T: 25 °C; t:NR | | 90% degradation efficiency | 2016 | [81] |
| NiO PZC = 8.45 + | Carbamazepine pK$_a$ = 13.9 + | O$_3$ dose: 5.5 g/m$^3$; Catalyst dose: 500 mg/L; pH: 3,4; T: 25 °C; t: 5 min | | 79.2% TOC removal | 2020 | [93] |
| CeO$_2$ PZC = 8.1 + | SMX pK$_a$ = 5.6–5.8 + | O$_3$ dose: 50 g/Nm$^3$; Catalyst dose: 100 mg; pH: 4.8; T: NR; t: 3 h | | 61% TOC removal | 2013 | [94] |
| γ-Al$_2$O$_3$ PZC = 8.3 + | 2,4 dimethylphenol pK$_a$ = 10.4 + | O$_3$ dose: 2 g/Nm$^3$; Catalyst dose: 5 g/L; pH: 4.5; T: 25 °C; t: 300 min | | 57% TOC removal | 2015 | [95] |
| γ-Al$_2$O$_3$ PZC = 8.3–8.9 + | Landfill leachate | O$_3$ dose: 22 mg/min; Catalyst dose: 50 g/L; pH: 7.3; T: NR; t: 30 min | | 70% COD removal | 2018 | [96] |
| Al$_2$O$_3$ PZC = 7.2–9.2 + | Textile wastewater | O$_3$ dose: 0.9 mmol/L; Catalyst dose: 300 g; pH: 4; T: NR; t: NR | | 25.83% COD removal | 2015 | [97] |

**Table 5.** *Cont.*

| Catalysts | Target Pollutants | Operating Conditions | | Removal Results | Year | Ref. |
|---|---|---|---|---|---|---|
| | | pH ≈ PZC | pH > pKa, pH ≈ pKa, pH < pKa | | | |
| NiCo$_2$O$_4$ PZC = 5 N | Sulfamethazine pKa = 2.65 − | O$_3$ dose: 4.5 mg/min; Catalyst dose: 0.05 g/L; pH: 5.2; T: NR; t: 60 min | | 34.1% TOC removal | 2021 | [78] |
| α-MnO$_2$ PZC = 3–5 N | 4-Nitrophenol pKa = 7.15 + | O$_3$ dose: 20 mg/L; Catalyst dose: 100 mg, pH 3.5–5.9; T: NR; t: NR | | 96.7% degradation 79.5% TOC removal | 2015 | [98] |
| | | pH > PZC | pH > pKa, pH ≈ pKa, pH < pKa | | | |
| α-Al$_2$O$_3$ PZC = 4.2 − | Humic acids pKa = 2.5 − | O$_3$ dose: 0.063 m$^3$/h; Catalyst dose: 0.5 g/L; pH: 5.5; T: 25 °C; t: 1 h | | 100% Humic acid removal | 2020 | [74] |
| γ-Al$_2$O$_3$ PZC = 8.3–8.9 − | CPF pKa = 6.38 − | O$_3$ dose: 1.4 mg/L.min; Catalyst dose: 0.55 g/L; pH: 9.5; t: 60 min | | 93% removal efficiency | 2019 | [99] |
| Ca$_2$Fe$_2$O$_5$ PZC = 9.5 − | Quinoline | O$_3$ dose: 17 mg/L; Catalyst dose: 1 g; pH: 10.5; T: 25 °C; t: 120 min | | 92% COD removal | 2022 | [100] |
| CeO$_2$–MnO$_2$ PZC = 10.13 − | Ammonium pKa = 9.25 − | O$_3$ dose: 12 mg/min; Catalyst dose: 1 g/L; pH: 11; T: NR; t: 60 min | | 88.14% Ammonium removal | 2022 | [101] |
| γ-Al$_2$O$_3$ PZC = 8.3–8.9 − | Naphthenic acid pKa = 5–6 − | O$_3$ dose: NR; Catalyst dose: 1 g/L; pH: 8.5; T: 25 °C; t: 50 min | | 88% Naphthenic acids removal | 2019 | [102] |
| CuFe$_2$O$_4$ PZC = 6–7 − | Phenacetin pKa = 2.2 − | O$_3$ dose: 0.36 mg/L; Catalyst dose: 2.0 g/L; pH: 7.72; T: NR; t: 30 min | | 95% of degradation | 2015 | [53] |
| α-MnO$_2$ PZC = 3–5 − | BPA pKa = 10.29 + | O$_3$ dose: 4.47 mmol/min; Catalyst 0.1 mg/L; pH: 6.25; T: 20 °C; t: NR | | 93.5% removal efficiency | 2015 | [103] |

NR—value not reported, TOC—total organic carbon, COD—chemical oxygen demand.

In Table 5, the PZC values of the metal oxide catalysts with the pKa of the pollutants in different operating conditions are correlated in order to identify the most favorable combinations for their removal.

From this comparison, it can be seen that the best removal efficiency trend is provided for experimental pH values in which the catalyst surface is charged. The electrostatic repulsion between pollutant and catalyst allows rapid O$_3$ decomposition, as well as the formation of radical species that occurs in the proximity of the pollutant adsorbed on the surface of the catalyst, seem to be the most effective mechanisms.

Al$_2$O$_3$ is one of the most popular materials in the catalytic ozonation process, among which several studies performed at pH close to neutral confirm it as a catalyst with excellent removal yields. The PZC value of Al$_2$O$_3$ can be different due to the catalyst's various impurities content, synthesis route, or thermal history, but the approximate range of PZC value is 7.2–9.4 [75,90,97]. Scrutinizing the study of Ziylan-Yavaş et al. [90] that had more than 95% removal of PCT (pKa ≈ 9.5 (Table 3)) by using γ-Al$_2$O$_3$ (PZC = 8.3–8.9 (Table 4)) as a catalyst verified that the optimal condition was when pH of the solution was lower than pKa and PZC. In these conditions, both catalyst and pollutant are positively charged. So, the governing mechanism was the repulsive electrostatic forces resulting in the desorption of the micropollutant from the catalyst which does not occupy the active surface sites, favoring the O$_3$ adsorption and its decomposition. Nemati Sani et al. [99] studied the catalytic efficiency of Al$_2$O$_3$ for CPF degradation in the catalytic ozonation process. Based on their work, the highest removal efficiency was at pH = 9.5. In this condition PZC < pH, pK$_a$ < pH, which means both catalyst and pollutant are negatively charged. So, as previously mentioned, in this condition there is not any effective adsorption of pollutants on the surface of the catalyst, and the primary mechanism is related to O$_3$ decomposition. In explaining their work, they clearly mentioned that ozonation is responsible for 88% of the CPF removal efficiency, which is another proof of the accurate understanding of govern mechanism categories in this review.

MgO is another efficient metal oxide that had excellent results in micropollutants degradation. The PZC value of MgO is in the range of 12–13 approximately [83]. With this PZC in a wide range of solution pH, MgO is positively charged. So oxidation of micropollutants due to the generation of $HO^\bullet$ radicals in the solution is the governing mechanism for catalytic ozonation by using this metal oxide. Based on using MgO as a catalyst, several works with considerable target pollutants removal are reported. Mashayekh-Salehi et al. [16] achieved complete degradation of acetaminophen in only 10 min of reaction time in the catalytic ozonation process. Based on our categorization, this work is in $pH < PZC$, $pH < pK_a$ condition. So, as previously mentioned, there is no effective adsorption of pollutants on the catalyst's surface due to the similar charges (positive charges). The primary mechanism is related to the oxidation of micropollutants due to the generation of $HO^\bullet$ radicals in the solution. It is worth mentioning, the authors confirmed that the reaction with $HO^\bullet$ radical was the leading cause of ACT oxidation using the $MgO/O_3$ process. The same condition ($pH < PZC$, $pH < pK_a$) was applied by Moussavi et al. [88], Kong et al. [17], and Mousavi et al. [91]. The exciting results of comparing these works was that they all achieved more than 95% degradation of their target pollutant, which is almost the top result for the catalytic ozonation process. This observation may prove that MgO is one of the best catalysts for the ozonation process, and this condition ($pH < PZC$, $pH < pK_a$) may be one of the best catalytic ozonation conditions, especially for the removal of very weak acid pollutants.

Furthermore, some scientists indicated that the catalytic mechanism and removal efficiency highly depend on the catalyst's characteristics. In this regard, introducing bimetal/polymetal oxides and using different synthesis routes with better chemical and physical properties was another way of improving this field. The proposed catalytic ozonation mechanisms are also rationalized for these kinds of catalysts. Oputu et al. [89] studied catalytic ozonation activity using β-FeOOH as a catalyst and 4-chloro phenol as the target pollutant. The pH of the solution was 3.5, so based on reported PZC (Table 4) and pKa (Table 3), both the pollutant and catalyst surface was a positive charge. According to the governing mechanism, the $O_3$ adsorption then its decomposition are favored, and there is no adsorption of pollutants on the surface due to the same positive charges. The interesting results based on this work were that in the presence of $O_3$ and catalyst, the removal efficiency was 99% (almost complete removal); however, in the absence of $O_3$, using only catalyst, the removal efficiency was 3%. This thought-provoking achievement shows that $O_3$ decomposition is the main effecting parameter in this condition; furthermore, the effect of pollutant adsorption on the catalyst surface is negligible.

Other metal oxide catalysts need to be mentioned, such as magnetic $Fe_3O_4$ nanoparticles [104] and δ-$MnO_2$ [105]. These metal oxides have shown good removal efficiencies but have not been reported in Table 5 as their pH values are not reported in the articles so their unclear process conditions do not allow to frame them in the proposed mechanism.

After perusing a lot of published articles from previous years, it is important to highlight that due to a large number of studies on these materials and the development of our understanding and knowledge about them, these materials can now be used for real wastewater plants. By scrutinizing the trend of published articles, changes in applications from laboratory scale to real wastewater plant can be observed.

### 4.2. Metal/Metal Oxides on Supports

This kind of catalyst was prepared by loading metal or metal oxides on supporters with unique surface properties due to increasing the catalytic activity of metal/metal oxides in the ozonation process. By applying this kind of material, both the surface area and the active sites of the materials would be increased. By the combination of metals/metal oxides and supporters, the PZC value of the prepared catalyst shifts to more acidic/basic, resulting in a new PZC value. This PZC change is attributed to the complexation of loaded metal/metal oxides onto the support surface. So, the kind of loading materials is important in this case. The following will clarify the governing mechanisms by scrutinizing several

related articles in this field. Table 6 systemizes the literature that employed metal/metal oxides as supports for the degradation of pollutants in the catalytic ozonation process. Operating conditions (pH, $O_3$ dosage, catalyst dose, etc.), the catalysts used, target pollutants, and removal results are reported.

**Table 6.** Literature reports on different metals/metal oxides on support as catalysts in the ozonation process (see Figures 3–5, respectively).

| Catalysts | Target Pollutants | Operating Conditions | Removal Results | Year | Ref. |
|---|---|---|---|---|---|
| | | pH < PZC      pH > pK$_a$, pH ≈ pK$_a$, pH < pK$_a$ | | | |
| Ce$_{1.0}$Fe$_{0.9}$OOH<br>PZC = 7.8<br>+ | Sulphamethazine<br>pKa = 2.65<br>− | $O_3$ dose: 15 mg/L; Catalyst dose: 0.2 g/L; pH 7.3; T: NR; t: 15 min | 41.2% mineralization efficiency | 2016 | [106] |
| LaTi$_{0.15}$Cu$_{0.85}$O$_3$<br>PZC = 9.8<br>+ | SMX<br>pKa = 5.6–5.8<br>− | $O_3$ dose: 25 mg/L; Catalyst dose: NR; pH: 7; T: 20 °C; t: 2 h | 85% TOC removal | 2009 | [107] |
| Ce deposited magnetic pyrite cinder<br>PZC = 7.63<br>+ | Reactive black-5<br>pKa = 3.8<br>− | $O_3$ dose 5.6 mg/min; Catalyst dose: 2.5 g/L; pH: 5.5; T: NR; t: 2 h | 83.32% TOC removal efficiency | 2016 | [69] |
| Ca-C/Al$_2$O$_3$<br>PZC = 9.53<br>+ | High-salt organic wastewater | $O_3$ dose: 12 mg/L; Catalyst dose: 20 g; pH: 8.36; T: NR; t: 40 min | 64.4% COD removal efficiency | 2022 | [108] |
| Mn-CeOx@γ-Al$_2$O$_3$<br>PZC = 9.37<br>+ | CPF<br>pKa = 6.38<br>− | $O_3$ dose: 13.969 ± 0.434 mg/L; Catalyst dose: 80 g/L; pH: 8.5; T: NR; t: 60 min | 100% CPF removal | 2022 | [109] |
| Mg-doped ZnO<br>PZC = 11–11.2<br>+ | Isoniazid<br>pKa = 1.82 | $O_3$ dose: 10–25 mg/L; Catalyst dose: 0.1 g/L; pH: 7.2; T: NR; t: 9 min | 76.3% removal efficiency | 2020 | [49] |
| MnOx/SBA-15<br>PZC = 4.27–6.35 | Clofibric acid<br>pKa = 3.2 | $O_3$ dose: 1 mg/L; Catalyst dose: 0.2 g/L; pH: 3.85; T: 293 K; t: 15 min | 43.8% TOC removal | 2015 | [20] |
| Fe-SBA-15<br>PZC = 4.0<br>+ | OA<br>pKa1 = 1.14; pKa2 = 3.64<br>N/− | $O_3$ dose: 100 mg/h; Catalyst dose: 0.24 g; pH: 3; T: NR; t: 60 min | 86.6% removal efficiency | 2016 | [21] |
| Fe-MCM-41<br>PZC = 4.95<br>+ | OA<br>pKa1 = 1.25; pKa2 = 3.81<br>N/− | $O_3$ dose: 21.8 mg/L; Catalyst dose: 0.4 g; pH: 3.6; T: NR; t: 30 min | 94% degradation<br>6% TOC/TOC$_o$ reduction | 2017 | [110] |
| SnOx-MnOx@Al$_2$O$_3$<br>PZC = 8.7<br>+ | PH<br>pKa = 9.98<br>+ | $O_3$ dose: 6 mg/L.min; Catalyst dose: 40 g/L; pH: 7; T: 20 ± 5 °C; t: 240 min | 93.8% COD removal efficiency | 2022 | [111] |
| | | pH ≈ PZC      pH > pK$_a$, pH ≈ pK$_a$, pH < pK$_a$ | | | |
| 4%Mn/γ-Al$_2$O$_3$<br>PZC = 6.75<br>N | PH<br>pKa = 9.98<br>+ | $O_3$ dose: 8 mg/min; Catalyst dose: 0.20 g; pH: 6.5; T:15 °C; t: NR | 82.67%<br>degradation efficiency | 2016 | [18] |
| MnOx-0.013/KCC-1<br>PZC = 4.0<br>N | OA<br>pKa1 = 1.14; pKa2 = 3.64<br>− | $O_3$ dose: 20 mg/L; Catalyst dose: 0.25 g/L; pH: 3.8; T: 25 °C; t: NR | 86% TOC removal | 2016 | [112] |
| γ-Ti-Al$_2$O$_3$<br>PZC = 7.3<br>N | SMX<br>pKa = 5.6–5.8<br>− | $O_3$ dose: 30 mg/Nm$^3$; Catalyst dose: 1.5 g; pH: 7; T: NR; t: 1 h | 92% TOC removal | 2017 | [113] |
| | | pH > PZC      pH > pK$_a$, pH ≈ pK$_a$, pH < pK$_a$ | | | |
| Cu–O–Mn/γ-Al$_2$O$_3$<br>(CMA)<br>PZC = 7.9<br>− | Polyvinyl alcohol<br>pKa = 5–6.5<br>− | $O_3$ dose: 5.52 mg/L.min; Catalyst dose: 150 mg/L; pH: 10, T:25 °C; t: 10 min | 99.3% PVA removal | 2020 | [19] |
| Fe-MCM-41<br>PZC = 4.85<br>− | Diclofenac<br>pKa = 4.15<br>− | $O_3$ dose: 100 mg/h; Catalyst dose: 1 g/L; pH: 7; T: NR; t: 30 min | 76.3% mineralization<br>70% TOC reduction | 2016 | [114] |
| Cu/Al$_2$O$_3$ | Herbicide Alachlor | $O_3$ dose: 12.2 mg/L.min; Catalyst dose: 0.27 g/L; pH: 4.4; T: 20 °C; t:NR | 75% TOC removal | 2013 | [115] |
| FeMn-MCM-41 | methyl orange | $O_3$ dose: 35 mg/L; Catalyst dose: 0.2 g; pH: 7; T: 25 °C; t:NR | 78% TOC removal | 2021 | [116] |
| MnOx/SBA-15<br>PZC = 5.33–6.06 | Norfloxacin | $O_3$ dose: 100 mg/h; Catalyst dose: 0.1 g/L, Catalyst loading 2%; pH: 5; T: 298 K; t: 60 min | 54% mineralization efficiency | 2017 | [117] |

**Table 6.** *Cont.*

| Catalysts | Target Pollutants | Operating Conditions | Removal Results | Year | Ref. |
|---|---|---|---|---|---|
| $MnO_2/Al_2O_3$ | Quinoline | $O_3$ dose: 135 mg/L; Catalyst dose: 70 g/L, 8% $MnO_2$ loading; pH: NR; T: NR; t: 90 min | 95% quinoline removal 65% TOC removal | 2021 | [118] |
| Fe silicate-loaded pumice | Diclofenac pKa = 4.15 | $O_3$ dose: 5 g/L; Catalyst dose: 8 g/L; pH 5; t: 30 min | 73.3% mineralization 21.17% TOC removal | 2017 | [119] |
| MOF-derived $Co_3O_4$–C composite | Norfloxacin | $O_3$ dose: 15 mg/L; Catalyst dose: 0.05 g/L; pH: 6.7; T: NR; t: NR | 48% TOC removal | 2021 | [25] |
| Fe silicate doped FeOOH | p-Chloronitro benzene | $O_3$ dose: 0.6 mg/L; Catalyst dose: 500 g/L; pH: 7; T: NR; t: 15 min | 61.3% TOC removal | 2017 | [120] |
| Ni/NHPC | linear alkylbenzene sulfonate | $O_3$ dose: NR; Catalyst dose: 0.3; pH: 10; T: NR; t: 30 min | 98.1% LAS removal | 2021 | [121] |
| $CuO/SiO_2$ | Oxalate | $O_3$ dose: 4 mg/L; Catalyst dose: 0.5 g, 4.5% metal loading; pH: 7; T: NR; t: NR | 95% oxalate removal | 2019 | [122] |
| $Pt-Al_2O_3$ | PCT pKa = 9.4–9.5 | $O_3$ dose: 6 mg/L; Catalyst dose: 5 mg/L; pH: 7; T: NR; t: 5 min | 100% PCT removal | 2018 | [90] |
| $MnO_2/CeO_2$ | Sulfosalicylic acid | $O_3$ dose: 4 mg/min; Catalyst dose: 0.1 g; pH: 6.5; T: NR; t: 30 min | 97% TOC removal | 2016 | [123] |
| $Fe_3O_4/Co_3O_4$ | SMX pKa = 5.6–5.8 | $O_3$ dose: 6 mg/L; Catalyst dose: 0.1 g/L; pH: 5.1; T: 25 °C; t: NR | 60% TOC | 2019 | [124] |
| Mn-Ce-O | SMX pKa = 5.6–5.8 | $O_3$ dose: 120 mg/h; Catalyst dose: 1 g/L; pH: 6.9; T: NR; t:120 min | 69% COD removal | 2015 | [125] |
| $CuO/Al_2O_3$-EPC | SMX pKa = 5.6–5.8 | $O_3$ dose: 4.75 µM; Catalyst dose: 0.5 g/L; pH: 6.2; T: 20 °C; t: 15 min | 87% SMX removal 21.2% TOC removal | 2019 | [126] |
| $Fe^{2+}$-Montmorillonite | SMX pKa = 5.6–5.8 | $O_3$ dose: 5 mg/min; Catalyst dose: 1 g/L; pH: 2.88; T: NR; t: 20 min | 97% COD removal | 2015 | [127] |

NR—value not reported, TOC—total organic carbon, COD—chemical oxygen demand.

Wang et al. [18] studied the enhancement of PH removal in the catalytic ozonation process by introducing an Mn-doped $Al_2O_3$ nanocatalyst. Based on the Mn weight ratios (2 wt.% Mn/$\gamma$-$Al_2O_3$, 4 wt.% Mn/$\gamma$-$Al_2O_3$, 8 wt.% Mn/$\gamma$-$Al_2O_3$), the PZC values were measured. The study showed that by increasing the amount of loaded Mn, the PZC values of the catalyst decreased from 7.31 to 6.75 to 5.54, respectively. Studying the effect of Mn loaded, they observed that at the natural pH (pH = 6.5), 4 wt.% Mn/$\gamma$-$Al_2O_3$ showed the best efficiency on PH (pKa $\approx$ 9.98 (Table 3)) removal. It means $O_3$ decomposition is happening in the environment according to our suggested mechanism. Yan et al. [19] used $\gamma$-$Al_2O_3$ support but with another modification of the surface. In this study, Cu–O–Mn/$\gamma$-$Al_2O_3$ was used as a catalyst in the ozonation system for the degradation of polyvinyl alcohol. Based on their investigation related to PZC measurement, the PZC of $\gamma$-$Al_2O_3$ decreased from 8.4 to 7.9 after loading Cu and Mn. They mentioned that the optimal condition is when the PZC of the catalyst and pH of the solution are the same or the pH is more than PZC. In these conditions, the governing mechanism is related to the existing hydroxyl groups on the surface of the catalyst, the decomposition of $O_3$, and the generation of reactive oxidation species. In another work, Bing et al. [113] studied the mineralization of pharmaceuticals over $\gamma$-Ti-$Al_2O_3$ catalyst. The exciting part related to this article was scanning the surface reaction mechanism. Using in situ Raman spectroscopy, they characterized intermediate species formed on the $\gamma$-Ti-$Al_2O_3$ surface with an aqueous $O_3$ solution. They observed the appearance of two new peaks at 880 and 930 cm$^{-1}$ in $\gamma$-Ti-$Al_2O_3$ suspension with $O_3$, which were related to surface peroxide ($O_2^{\bullet}$) and surface atomic oxygen species ($O^{\bullet}$), respectively. As mentioned before in the governing mechanism explanation, these species would generate when the PZC of the catalyst is the same as the pH of the solution. They reported that the PZC value for this synthesized catalyst was 7.3 and the pH of the solution was 7. This work is an excellent observation for reliability even the details on intermediate species of our proposed govern mechanism.

One of the prominent supporting materials for catalysts is silica-based materials (such as $SiO_2$, SBA-15, MCM-41, etc.) due to their large surface area, good flexibility, stability,

adjustable structure, and biocompatibility. The following examples comprehensively illustrate the diversity of using these materials in catalytic ozonation.

Jeirani et al. [110] worked on a modified mesoporous Fe-MCM-41 catalyst to remove OA as a target pollutant. Their evaluation of the adsorption and ozonation process and the determination of PZC and pKa was thought-provoking. They specifically describe the dissociation of OA in water. OA (target pollutant) was ionized to hydrogen-oxalate anion ($C_2O_4H^-$), having a negative charge on the surface. On the other hand, the PZC value for Fe-MCM-41 and Mn, Ce/Fe-MCM-41 were measured (4.95 and 6, respectively); accordingly, the catalyst's surface was positively charged (pH < PZC). Then by comparing the adsorption and ozonation efficiency of the catalyst, they observed that adsorption was the governing mechanism for this treatment system, which is in line with our previous explanation of the mechanism.

Yan et al. [21] applied another silica-based material for OA removal. They found Fe-doped SBA-15 (PZC = 4) as a potential catalyst for the catalytic ozonation process. They studied the influence of initial pH in the range of 1 to 9 on the removal process. Based on their results and the known information about the pKa value of OA and PZC value of Fe-SBA-15, the best pH was 3 with 97.4% OA removal efficiency. This achievement affirms that the negatively charged micropollutants adsorbed on the positively charged catalyst surface. Thus, the contaminants were near where the generation of $^\bullet$OH radicals happens, which means $^\bullet$OH can quickly oxidize them in the environment. This mechanism is similar to the previous example of OA removal from wastewater.

Shen et al. [49] studied a new Mg-doped ZnO catalyst. By modifying the catalyst surface, the value of PZC was changed from 8 for ZnO support to 11.2 for the Mg-ZnO catalyst. Furthermore, by increasing the doping amount of Mg, the PZC value was increased. Interestingly, by changing the pH value from 3 to 9, the efficiency of the catalyst did not noticeably change because all of those pH values were less than the PZC of the catalyst (pH < PZC), so the reaction mechanism for all of those conditions were the same. Furthermore, their explanations related to the catalytic mechanism, surface charge, and kind of active radicals were another validation of our suggested governing mechanism.

For many of these supported catalysts, the lack of studies on the new PZC values makes it difficult to predict their mechanism of action, although they show high removal efficiency as reported in Table 6.

### 4.3. Carbon-Based Materials

Applying carbon-based materials has received significant popularity in the catalytic ozonation process during past decades. The variety of these materials, their good catalytic performance, and their environmentally friendly properties make them very attractive for these studies. Bulky carbons (such as activated carbon (AC)), nano carbons (such as carbon nanotubes (CNTs) and graphene), and carbon-based nanocomposites are the three most popular materials in catalytic ozonation. Table 7 compiles the literature results employing carbon-based materials for the degradation of organics. As can be seen, the systems have been investigated over a wide range of operating conditions (pH, $O_3$ dosage, the mass ratio between solid and organic matter load, etc.), the kind of used catalysts, target pollutants, and removal results. The articles for summarizing this part were chosen according to their high citations.

**Table 7.** Literature reports on different carbon-based materials as catalysts in the ozonation process (see Figures 3–5, respectively).

| Catalysts | Target Pollutants | Operating Conditions | Removal Results | Year | Ref. |
|---|---|---|---|---|---|
| | | pH < PZC          pH > $pK_a$, pH ≈ $pK_a$, pH < $pK_a$ | | | |
| Fe/AC PZC = 7.95 + | Crystal violet dye pKa1 = 1.15 pKa2 = 1.8 − | $O_3$ dose: 4.44 mg/min; Catalyst dose: 2.5 g/L; pH: 7; T: NR; t: NR | >96% decolorization | 2015 | [128] |
| AC PZC = 8.5 + | SMX pKa = 5.6–5.8 + | $O_3$ dose: 50 g/Nm$^3$; Catalyst dose: 100 mg; pH: 4.8; T: NR; t: 3 h | 45% TOC removal | 2011 | [129] |
| AC PZC = 8.5 + | SMX pKa = 5.6–5.8 + | $O_3$ dose: 48 mg/L; Catalyst dose: 2 g/L; pH: 5; T: 26 °C; t: 20 min | 78% TOC removal | 2011 | [130] |
| Treated Commercial MWCNT-HNO$_3$-N$_2$-900 PZC = 7.3 + | SMX pKa = 5.6–5.8 + | $O_3$ dose: 50 g/Nm$^3$; Catalyst dose: 100 mg; pH: 4.8; T: NR; t: 3 h | 45% TOC removal | 2013 | [131] |
| Treated Commercial MWCNT-O$_2$ PZC = 5.2 + | SMX pKa = 5.6–5.8 + | $O_3$ dose: 50 g/Nm$^3$; Catalyst dose: 100 mg; pH: 4.8; T: NR; t: 3 h | 41% TOC removal | 2013 | [131] |
| MWCNT PZC = 7 + | SMX pKa = 5.6–5.8 + | $O_3$ dose: 50 g/Nm$^3$; Catalyst dose: 100 mg; pH: 4.8; T: NR; t: 3 h | 35% TOC removal | 2011 | [130] |
| AC$_0$ PZC = 8.5 + | RB5 | $O_3$ dose: 50 g/Nm$^3$; Catalyst dose: 350 mg; pH: 5.6; T: 25 °C; t: 2 h | 70% TOC removal | 2009 | [86] |
| AC$_0$-Ce-O composite PZC = 8.5 + | RB5 | $O_3$ dose: 50 g/Nm$^3$; Catalyst dose: 350 mg; pH: 5.6; T: 25 °C; t: 2 h | 100% TOC removal | 2009 | [86] |
| | | pH ≈ PZC          pH > $pK_a$, pH ≈ $pK_a$, pH < $pK_a$ | | | |
| MnOx/sewage sludge-derived AC PZC = 3.5 N | OA pKa1 = 1.14; pKa2 = 3.64 N/− | $O_3$ dose: 5 mg/L; Catalyst dose: 100 mg/L, Catalyst loading 30%; pH: 3.5; T: NR; t: 60 min | 92.2% removal efficiency | 2017 | [132] |
| Fe-MnO$_X$/AC PZC = 6.1 N | PH pKa = 9.9 + | $O_3$ dose: 60 mg/L; Catalyst dose: 1 g/L; pH: 6; T: NR; t: 20 min | 90.75% TOC removal | 2022 | [133] |
| | | pH > PZC          pH > $pK_a$, pH ≈ $pK_a$, pH < $pK_a$ | | | |
| AC Darco 12–20 PZC = 6.4 − | SMX pKa = 5.6–5.8 − | $O_3$ dose: 25 mg/L; Catalyst dose: NR; pH: 7; T: 20 °C; t: 2 h | 92% TOC removal | 2012 | [107] |
| AC/nano-Fe$_3$O$_4$ PZC = 6.08–7.7 − | PH pKa = 9.9 + | $O_3$ dose: 33 mg/L.min; Catalyst dose: 2 g/L; pH: 8; T: NR; t: 60 min | 98.5% PH removal 69.8% COD removal | 2014 | [134] |
| CeO$_2$/MWCNT | SMX | $O_3$ dose: 50 g/Nm$^3$; Catalyst dose: 100 mg; pH: 4.8; T: NR; t: 3 h | 56% TOC removal | 2013 | [94] |
| Fe$_2$O$_3$/CeO$_2$ loaded AC (MOPAC) | SMX | $O_3$ dose: 48 mg/L; Catalyst dose: 2 g/L; pH: 5; T: 26 °C; t: 20 min | 86% TOC removal | 2011 | [130] |
| rGO | p-Hydroxylbenzoic Acid (PHBA) | $O_3$ dose: 20 mg/L; Catalyst dose: 0.1 g/L mg; pH: 3.5; T: 25 °C; t: 30 min | 100% PHBA removal | 2016 | [135] |
| α-MnO$_2$/RGO | BPA | $O_3$ dose: 4.47 mmol/min; Catalyst dose: 0.1 mg/L; pH: 6.25; T: 20 °C; t: 60 min | 90.5% BPA removal | 2015 | [103] |
| GO/Fe$_3$O$_4$ | ρ-chlorobenzoic acid (pCBA) | $O_3$ dose: 4 mg/L; Catalyst dose: 20 mg/L; pH: 7; T: NR; t: 5 min | 51% TOC removal | 2018 | [136] |
| Heteroatom doped graphene oxide PGO | SMX | $O_3$ dose: 2 g/h; Catalyst dose: 1 g/L; pH: 9; T: 25 °C; t: 5 min | 99% SMX removal | 2017 | [137] |

NR—value not reported, TOC—total organic carbon, COD—chemical oxygen demand.

As it is apparent in Table 7, AC or carbon black is one of the most used catalysts for the ozonation process. High porosity, surface functionalities, its low cost are the reasons for its extensive utilization. The PZC value of AC can be different due to the catalyst's various impurities content, synthesis route, or thermal history, and the method for investigation of PZC, while the reported range of PZC value for AC is between 4.9–11.9 [68,138]. On the other hand, commercially available AC can be modified by minerals such as alkali metals

(Ca, Na, K, Li, Mg) or multivalent metals (Al, Fe, Ti, Si) and metal oxides. The presence of impurities on the surface of AC would significantly affect the PZC values. In most articles, the authors reported this value for the AC used in their work.

Shahamat et al. [134] studied a new carbon-based catalyst called AC/nano-$Fe_3O_4$ to remove PH. During this study, they calculated the PZC of the catalyst and illustrated that when the pH of the solution is between PZC and pKa, the negative catalytic charge and positive charge of the pollutant can attract each other on the surface of the catalyst. $O_3$ decomposition is the primary reaction mechanism when the catalyst has a negative charge. These conditions were responsible for achieving the optimal efficiency for PH removal. In another work, Huang et al. [132] synthesized MnOx/sewage sludge-derived AC (MnOx/SAC) to improve the catalytic efficiency of OA degradation in ozonation. Based on the report, the best organic contaminant removal was at the pH equal to the PZC of the catalyst (PZC = 3.5), and the governing mechanism was related to existing hydroxyl radicals on the catalyst's surface which is the starter part for $O_3$ decomposition. Synthesized Fe-loaded AC for dibutyl phthalate removal was another work by Huang et al. [139], which had the same result that an uncharged surface with hydroxyl radicals on the surface was more active than the charged surface.

CNTs and MWCNTs are used frequently in the catalytic ozonation process as the mixed mesoporous structured nanocarbons. Acceleration of reaction kinetics, rapid mass diffusion, large surface area, and facile modification of surface are the main advantages of this kind of catalyst. Several techniques were used to promote the catalytic activity of this material, such as substituting carbon atoms with metal-free heteroatoms (e.g., N, S, and F).

Gonçalves et al. [131] studied the effect of MWCNTs on the catalytic ozonation of SMX (pKa ≈ 5.6–5.8 (Table 3)). A set of modified MWCNTs with different levels of acidity/basicity was prepared and tested. The PZC value of the original MWCNT was 7; however, by modification of the original catalyst, the amount of PZC was changed to more acidic and basic. Based on the results, all those catalysts illustrated excellent efficiency for SMX removal, but one of the modified MWCNTs, called MWCNT-$HNO_3$_$N_2$_900 with PZC 7.3, illustrated better catalytic efficiency than the others. Based on our categorization and the observation in this work, at pH < PZC, pH < pKa condition, there is no effective adsorption of pollutants on the catalyst's surface due to the similar charges (positive charges). So the primary mechanism is related to the oxidation of micropollutants due to the generation of $HO^\bullet$ radicals in the solution.

Graphene oxide (GO) and reduced graphene oxide (rGO) are other prominent catalysts that have been extensively employed to accelerate the degradation of various contaminants by $O_3$. Scrutinizing the study of Wang et al. [135] that had complete PHBA removal (pKa ≈ 4.85) by using rGO (PZC = 4.7) as a catalyst verified that the optimal condition was at the pH of 3.5. As mentioned before, in the condition that pH < PZC, pH < pKa, both catalyst and pollutant are positively charged, which leads to no adsorption on the system so that the primary mechanism would be related to the generation of $HO^\bullet$ by $O_3$ decomposition.

### 4.4. Metal–Organic Frameworks (MOFs)

As a rapidly emerging category of porous materials, metal–organic frameworks (MOFs) are widely used in different research fields due to their unique topology, adjustable features, large surface area, ultrahigh porosity, and ease of access to numerous functional groups [27,140–143]. The presence of hydroxyl groups on the surface of MOFs and the open metal sites of MOFs are two powerful catalytic active sites for the ozonation process. Their presence plays an important role in the adsorption and decomposition of $O_3$. The catalytic efficiency of MOF highly depends on the type of metal incorporated in the MOF. Therefore, there are several studies on designing and synthesizing MOFs to produce an appropriate catalyst to be used in the catalytic ozonation process [26,121,143–149]. In recent years, several studies have been reported on the applicability of MOFs in the catalytic ozonation process, including Co/Ni-MOF [150], Ce-doped MIL-88A(Fe) [26], and Fe-based

MOFs [151], and this emerging category of materials could be one of high-potential materials by considering some improvements in the future.

## 5. Conclusions

This review focused on a bibliometric study of catalytic ozonation as one of the popular AOPs methods, conducted from 2000–2021. Nearly 600 articles published during this period, identified by the Web of Science (WOS) database and a bibliometric analysis using VOS viewer software, have been carefully examined and evaluated in terms of future development and practicability. The most impacting articles have been scrutinized and discussed in terms of the interpretation of mechanism and a new vision outlined on the evaluation of both heterogeneously and homogenously catalyzed ozonation processes for the degradation and mineralization of various toxic organic pollutants in water.

Particular attention has been devoted to describing the activities and efficiency of heterogeneous catalysts in the ozonation process related to the chemical properties of catalysts such as crystallographic and morphological, chemical stability as well as the opportune combination of their PZC values with pKa of the target pollutant and pH of the solution.

Examining the results related to the catalytic activity of the metal oxide catalysts, it can be emphasized that the best performance can be obtained when the PZC and pKa values produce positively charged catalyst surfaces and target pollutants, respectively. Despite the small number of citing works, the negatively charged pollutant and the catalyst surface seem a favorable combination for obtaining a good removal efficiency. At least for this type of heterogeneous catalyst, it can be assumed that the repulsion between the pollutants and catalysts promotes the formation of HO$^\bullet$ as the species responsible for the enhancement of the removal processes of the target pollutant. Carbon-based catalysts do not seem to follow this trend; for this reason, deeper investigations could be expected for this class of materials in the future.

Finally, we believe that this study may be of help to authors aiming to improve knowledge in this field.

**Author Contributions:** Conceptualization, N.F. and E.B.; methodology, N.F.; software, N.F.; validation, N.F., E.B., D.S. and G.M.; formal analysis, N.F. and E.B.; investigation, N.F. and E.B.; resources, N.F. and E.B.; data curation, N.F. and E.B.; writing—original draft preparation, N.F.; writing—review and editing, N.F., E.B. and G.M.; visualization, D.S.; supervision, D.S. and G.M; project administration, G.M.; funding acquisition, G.M. All authors have read and agreed to the published version of the manuscript.

**Funding:** PANIWATER project funded jointly by the European Union's Horizon 2020 research and innovation programme: 820718; Programma Operativo Nazionale Ricerca e Innovazione 2014–2020: F85F20000290007.

**Data Availability Statement:** Not applicable.

**Acknowledgments:** All individuals included in this section have consented to the acknowledgment.

**Conflicts of Interest:** The authors declare no conflict of interest.

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
