# Peer review of "State of Art and Perspectives in Catalytic Ozonation for Removal of Organic Pollutants in Water: Influence of Process and Operational Parameters"

_catalysts, doi:10.3390/catal13020324_

Round 1

Reviewer 1 Report

Recommendation: Published in catalysts after major revisions.

Comments:

Fallah and colleagues have focused on surveying and summarizing the application of catalytic ozone in water and wastewater treatment, especially for heterogeneous catalysts. This review using a combination of bibliometric analysis and systematic paper review presents a new perspective on assessing the activity of multiple heterogeneous catalysts by using chemical physical parameters such as point of zero charge (PZC), pKa and pH. It may be of help to authors aiming to improve knowledge in this field.

Q1.The quality of Figures 2, 3, 4 and 5 needs to be improved, mainly because the characters are too small to be seen.

Q2.More examples should be developed to support the view on the prospective inference of homogeneous catalyst transition metal ions as in situ catalysts for real wastewater.

Q3.Before assessing the effect of the relationship between pKa, pH and PZC on the catalytic activity, the general mechanism of action of the catalyst, pollutant and O3 in the catalytic process and the effect of pKa, pH and PZC on the process, respectively, should be briefly described in order to understand the subsequent discussion.

Q4.When summarizing the effect of the three parameters pKa, pH and PZC on the catalytic activity, how to exclude the effect of catalyst dosage, O3 dosage and other parameters? It is the complexity of the catalytic oxidation mechanism due to such various complex parameters.

Q5.The values of pKa for different pollutants and the range of PZC for selected catalysts are presented separately in Tables 3 and 4 and are not very useful and meaningful in the text.

Q6.The format of the tables in the text suggests the use of a conventional three-line diagram, which is more visual and aesthetically pleasing.

Q7.It is suggested to add a classification of the relationship between pKa, pH and PZC in Tables 5, 6 and 7 for a more visual discrimination.

Reviewer 2 Report

This manuscript aims to take a systematic review of the progress in catalytic ozonation for pollutant degradation, which is truly a hot topic in the field of environmental science and engineering. Catalytic ozonation is suggested as a powerful technique for treating various organic pollutants either by using homogeneous processes or heterogeneous processes. The latter is considered more technically available and useful in the engineering process. Unfortunately, some important progress in the heterogenous catalytic ozonation is missing here. Therefore, it needs a major revision before publication. Specific comments could be found below.

1. In the abstract and some parts of the manuscript elsewhere, it only assesses the point of zero charge (PZC) of catalysts is too narrow to give a big picture. Because except PZC, other properties like lewis acid sites are very important to initiate the catalytic ozonation, the authors must review this part of the mechanism and summarize the effect of Lewis acid sites in the manuscript.

2. In the introduction, catalytic ozonation process is less elucidated, which is unacceptable for the main topic of this manuscript--catalytic ozonation review. The authors should go follow this logic: why and how we develop catalytic ozonation, what are important aspects for catalyst design and practice, and how many catalysts we have now.

3. There has also been great progress in developing Heterogeneous catalysts for catalytic ozonation, including metal-based, metal-free, and metal-organic framework catalysts. In this manuscript, the authors only demonstrated a few of conventional catalysts, such as metal oxides. The authors must introduce some state-of-the-art catalysts, such as metal-organic frameworks (go for these works: Ozone: Science & Engineering, 43:3, 239-253, DOI: 10.1080/01919512.2020.1782725; Chinese Chemical Letters 33 (2022) 5013–5022; Journal of Hazardous Materials 403 (2021) 123697; Applied Surface Science 509 (2020) 145378; Applied Catalysis B: Environmental 251 (2019) 66–75; Chemosphere 291 (2022) 132874; Journal of Hazardous Materials, 2022, 431, 128575; Chemical Engineering Journal, 2021, 404, 127075; Nano Res. 2022, 15, 2961–2970; ). In addition, the active sites in catalysts responsible for catalyzing the decomposition of ozone into reactive oxygen species should be well-discussed rather than simply mentioning the previous reports, and some of these reports are not meaningful.

4. It is strongly suggested that more pictures of efficiency and mechanism studies should be added to draw more attention.

5. There are too many old papers cited in this manuscript and 2020-2022 years papers are very rare, and papers in important journals should have more citations. So this would make this manuscript out of the state-of-the-art

Round 2

Reviewer 1 Report

The manuscript has been modified as the reviewer's comment, it can be accepted as the present version.

Reviewer 2 Report

My concerns have been well addressed. I have no more comments.